# Lenvatinib for Hepatocellular Carcinoma: A Literature Review

**DOI:** 10.3390/ph14010036

**Published:** 2021-01-06

**Authors:** Takeshi Hatanaka, Atsushi Naganuma, Satoru Kakizaki

**Affiliations:** 1Department of Gastroenterology, Gunma Saiseikai Maebashi Hospital, 564-1 Kamishindenmachi, Maebashi, Gunma 371-0821, Japan; 2Department of Gastroenterology, National Hospital Organization Takasaki General Medical Center, 36 Takamatsucho, Takasaki, Gunma 370-0829, Japan; naganuma.atsushi.nj@mail.hosp.go.jp; 3Department of Clinical Research, National Hospital Organization Takasaki General Medical Center, 36 Takamatsucho, Takasaki, Gunma 370-0829, Japan; kakizaki@gunma-u.ac.jp; 4Department of Gastroenterology and Hepatology, Gunma University Graduate School of Medicine, 3-39-15 Showa, Maebashi, Gunma 371-8511, Japan

**Keywords:** lenvatinib, hepatocellular carcinoma, progression-free survival, overall survival, adverse events, post-progression treatment, BCLC intermediate stage, transcatheter arterial chemoembolization, nutrition assessment, sarcopenia

## Abstract

Lenvatinib, which is an oral multikinase inhibitor, showed non-inferiority to the sorafenib in terms of overall survival (OS) and a higher objective response rate (ORR) and better progression-free survival (PFS) in patients with hepatocellular carcinoma (HCC). A good liver function and Barcelona Clinic Liver Cancer (BCLC) intermediate stage were the key factors in achieving therapeutic efficacy. The management of adverse events plays an important role in continuing lenvatinib treatment. While sequential therapies contributed to prolonging overall survival, effective molecular targeted agents for the administration after lenvatinib have not been established. Repeated transcatheter arterial chemoembolization (TACE) was associated with a decline in the liver function and poor therapeutic response in BCLC intermediate patients. Recently, the Asia-Pacific Primary Liver Cancer Expert (APPLE) Consensus Statement proposed the criteria for TACE unsuitability. Upfront systemic therapy may be better for the BCLC intermediate stage HCC patients with a high tumor burden, while selective TACE will be recommended for obtaining a curative response in patients with a low tumor burden. This article reviews the therapeutic response, management of adverse events, post-progression treatment after Lenvatinib, and treatment strategy for BCLC intermediate stage HCC.

## 1. Introduction

Liver cancer is the sixth most frequent malignant tumor and fourth most common cause of cancer death worldwide, with 841,000 patients newly diagnosed and 782,000 cancer deaths annually [1]. Hepatocellular carcinoma (HCC) is the most frequent type of primary liver cancer, which accounts for approximately 80% of all cases [1]. Curative treatment, such as surgical resection, liver transplantation, and radiofrequency ablation, are indicated for patients with early stage HCC, and transcatheter arterial chemoembolization (TACE) is the standard treatment for the patients with intrahepatic multifocal HCC [2]. Systemic chemotherapy is one of the standard treatments for advanced HCC patients with well-preserved liver function [2]. In addition, systemic chemotherapy is also indicated for the patients with progression after TACE treatment [2], because repeated TACE reduces its therapeutic efficacy and worsens the liver function, which results in decreased survival time [3].

Sorafenib is an oral multikinase inhibitor of tyrosine kinase receptors, including vascular endothelial growth factor receptors (VEGFRs) and platelet-derived growth factor (PDGF) receptor-β, which is associated with peritumoral neovascularization and drivers of cell proliferation, such as RAF1, BRAF, and KIT [4]. Sorafenib showed a survival benefit in patients with advanced HCC, according to two pivotal phase 3 placebo-controlled studies, namely the SHARP trial [5] and the Asia-Pacific trial [6], which were reported in 2008 and 2009, respectively. Because randomized control trial (RCT) represents the gold standard in evaluating healthcare interventions [7], sorafenib is the first approved systemic agent for advanced HCC and it is widely used in clinical settings. However, the efficacy of sorafenib remains limited and unsatisfactory, showing that the median prolongation of overall survival was less than three months and the response rate was less than 3% [5,6].

Lenvatinib, which is an oral multikinase agent, acts as an inhibitor of VEGFR 1–3, fibroblast growth factor receptors 1–4, PDGF receptor-α, KIT, and RET [8,9,10]. The phase 2 study demonstrated a good therapeutic response and acceptable toxicity in patients with advanced HCC, with an objective response rate (ORR) of 37%, as assessed by the modified Response Evaluation Criteria in Solid Tumors (mRECIST), and 24%, as assessed by RECIST ver.1.1. The median time to progression (TTP) was 7.4 months and the median overall survival (OS) was 18.7 months [11]. The REFLECT trial, a global phase 3 study, was conducted in order to compare lenvatinib and sorafenib as 1st-line treatment. The key eligibility criteria for the REFLECT trial were no indication for surgical treatment, Barcelona Clinic Liver Cancer (BCLC) intermediate or advanced stage, Child-Pugh class A, and no prior systemic therapy. In the REFLECT trial, lenvatinib showed the non-inferiority to sorafenib in terms of OS (13.6 months vs. 12.3 months; hazard ratio (HR) 0.92) [12]. Lenvatinib also showed a higher ORR (40.6% by mRECIST and 18.8% by RECIST ver.1.1), better TTP (median, 7.4 months), and better progression-free survival (PFS; median, 7.3 months) [12]. Lenvatinib was recommended for the patients with BCLC advanced stage and patients with BCLC intermediate stage on progression after TACE, according to the American Association for the Study of Liver Diseases (AASLD) guideline 2018 [13]. The European Association for the Study of the Liver (EASL) also recommended the indication of lenvatinib for BCLC advanced stage (without tumor thrombosis at the main portal vein) and HCC that shows progression with or which is unsuitable for locoregional therapy in patients with Child-Pugh class A and good performance status (PS) [2].

Nowadays, lenvatinib is widely used for the treatment of advanced HCC. While real-world evidence is increasing, there are few reports on the comprehensive findings of lenvatinib treatment. In addition, many problems have remained unresolved. While regorafenib [14], cabozantinib [15], and ramucirumab [16] (limited to serum level of α-fetoprotein (AFP) ≥ 400 ng/mL) are established as treatments after progression on sorafenib, treatments that are effective after progression on lenvatinib have not been fully elucidated. While the efficacy and safety of lenvatinib for patients with BCLC intermediate stage is promising, the timing of the introduction of lenvatinib and switching from other treatments, including TACE, have not been fully discussed. The efficacy and safety of lenvatinib remains unknown for these patients because the REFLECT trial excluded patients who had received systemic therapy, and those with portal vein invasion at the main portal trunk, bile duct invasion, liver occupation of tumor ≥ 50% and Child-Pugh class B.

Thus, this present article reviewed the therapeutic response, the management of adverse events, post-progression therapy, and treatment strategy for the BCLC intermediate stage.

## 2. Therapeutic Response of Lenvatinib

### 2.1. The Ojective Response Rate of Lenvatinib

Many real-world studies have reported the short-term therapeutic response of lenvatinib, resulting in an ORR of 29.9–53.5%, as assessed by mRECIST [17,18,19,20,21,22,23,24], and 14–25%, as assessed by RECIST ver.1.1 [19,22,25,26]. These findings seemed to be in agreement with the results of phase 2 and 3 studies (Table 1). To date, many researchers have also investigated the ORR as a predictive factor in the clinical setting. Ueshima et al. [17] reported that the high ORR was found in patients with albumin-bilirubin (ALBI) grade 1 at baseline. We reported that BCLC intermediate stage was a significant predictive factor that was associated with the ORR in a multivariate analysis [18], which agreed with the results regarding the ORRs in the analysis of a Japanese subpopulation in the REFLECT trial [27]. A high ORR was found in patients with a good PS [21], Child-Pugh class A [21], and Child-Pugh score 5 (CP-5A) [23]. Some of the researchers also reported that the high relative dose intensity (RDI) at four weeks (30 days) [23,28,29] or eight weeks (60 days) [19,21,30,31] was relevant to therapeutic response of lenvatinib, including the PFS and OS (Table 2). The cut-off values of the RDI were similar, ranging from 66% to 75%. A high RDI was associated with a good liver function at baseline [19,28,29,30,31], low tumor stage [19,30], and an initial full dose of lenvatinib [19,30], which indicated that the liver function and early tumor stage would play an important role in obtaining a high RDI. Interestingly, low body weight [28] or BMI [19] were relevant to low RDI. While the reason for this is unclear, it might be related to the high incidence of adverse events (AEs) in patients with low BMI values [24]. In contrast, a retrospective study showed that the delivered dose intensity/body surface area ratio (DBR) was better a predictive factor than the RDI [31]. Because only one retrospective study has compared the RDI and DBR, further research was conducted in order to assess the usefulness of DBR. Some researchers reported the utility of contrast-enhanced ultrasonography at day 7 [32], AFP change at two weeks [33], and radiological imaging at two weeks [34] in order to predict the ORR of lenvatinib early, although the numbers of subjects were limited in these studies. These findings suggest that it is preferable to start lenvatinib at a full dose for patients with ALBI grade 1 and BCLC intermediate stage. When possible, an early tumor assessment was also preferable.

### 2.2. The Progression-Free Survival and Overall Survival

PFS was defined as the time from randomization (in phase 3 study) or start of treatment (in real-world studies) to disease progression or death. The median PFS in lenvatinib-treated patients was reported to be 7.3 months (95% confidence interval (CI) 5.6–7.5) in a phase 3 study [12], while some of the retrospective studies reported the median PFS of 4.3–9.8 months [17,18,20,22,23,25,26]. Our retrospective study reported that the BCLC intermediate stage was significantly associated with PFS in a multivariate analysis (median PFS; 8.0 months in intermediate stage, 4.0 months in advanced stage) [18]. There were no differences in PFS between the treatment-naïve patients and patients who had received previous treatment. The PFS in patients with extrahepatic spread was shorter in comparison to the patients without extrahepatic spread [18]. Another retrospective study showed that CP-5A, tumor size ≥ 40 mm were significant pretreatment factors and that the incidence of thyroid dysfunction and appetite loss was associated with worse PFS [23]. The results of the present study indicated that a good liver function, small tumor size, and careful management of AEs during treatment contributed to prolonged PFS [23]. Ono et al. reported that an RDI of >70% at four weeks was correlated with PFS [29]. Takahashi et al. revealed that patients with an RDI of ≥75% at eight weeks showed better PFS in comparison to those with RDI < 75% [30]. These studies indicated that a maintained RDI was associated with longer PFS [29,30]. Because the liver function and early tumor stage would play an important role in obtaining a high RDI, as we mentioned above, the administration of full-dose lenvatinib to patients with BCLC intermediate stage (or low tumor burden) would be expected to result in good PFS. The management of AEs was crucial in obtaining good PFS (Table 1).

The median OS was 13.6 months (95% CI 12.1–14.9) in a phase 3 study [12]. The median OS was shown to be 7.1–13.3 months, according to some retrospective studies [19,21,22,26], and modified ALBI (mALBI) grade 2b or 3 [24], Child-Pugh class B [21], BCLC advanced stage [21,35], and elevated C-reactive protein [36] were found to be significant unfavorable factors. While the starting dose of sorafenib did not influence OS according to one retrospective study [37], the patients with the higher RDI of lenvatinib (>67%) at eight weeks had significantly better OS in comparison to those with lower RDI (≤67%) [19]. However, care is required in the interpretation of the results of OS, because the observation period in many previous reports was insufficient. Accordingly, further research with an adequate follow-up period is warranted for investigating predictors of OS (Table 1).

The results of these real-world studies were mainly reported from Japan. According to the Japanese subpopulation analysis of the REFLECT trial [38], the Japanese patients were older, had a lower body weight, better PS, lower serum level of AFP, and, more frequently, had BCLC intermediate stage and underlying liver diseases of HCV in comparison to the overall population of the REFELCT trial. The OS of Japanese patients was better than that in the overall population of the REFLECT trial (17.6 months vs. 13.6 months), which may be explained by a low HCC stage, low serum level of AFP, and high percentage of patients receiving anticancer therapy after lenvatinib [38]. Hence, these tendencies might be found in these real-world studies and it might be necessary to pay attention to the interpretation of the results of the real-world studies.

## 3. Adverse Events

While physicians were accustomed to a sorafenib toxic profile that is based on ten-year experience, the experience in the management of lenvatinib toxicity was limited [39]. Therefore, we described not only the common adverse events (AEs), but also those requiring special attention in this paragraph.

### 3.1. The Fatigue, Appetite Loss and Treatment Discontinuation

Adverse events that were frequently reported in many previous studies included fatigue, appetite loss, hand-foot skin reaction (HFSR), diarrhea, hypertension, hypothyroidism, and proteinuria [12,18,21,22,23,24,40]. The rate of discontinuation due to AEs was 8.6–43.5% [18,22,40]. Thus, the management of AEs plays a central role in lenvatinib treatment. The median onset of fatigue, appetite loss, HFSR, and diarrhea was approximately one month after the start of lenvatinib, while the median onset of hypertension was on day 4 [40]. Among these frequent AEs, the occurrence of appetite loss was strongly correlated with the therapeutic effect [18,23,24]. In addition, fatigue and appetite loss were risk factors for treatment discontinuation [40]. Some of the studies showed that these AEs [18] and the incidence of discontinuation to due AEs [17,40] were less frequently found in patients with ALBI grade 1 in comparison to those with ALBI grade ≥2, indicating that AEs of lenvatinib were manageable in the patients with a good baseline liver function. In contrast, other studies reported that the frequency of appetite loss and fatigue was not significantly associated with the mALBI score [22,24]. One of the reasons was presumed to be the small number of the patients with a relatively poor liver function. Another reason was considered to be the shorter period of lenvatinib administration in patients with poorer liver function in comparison to those with a better liver function. Hence, the liver function will play a central role in the management of AEs.

Dose reduction or dose interruption are performed as countermeasures against fatigue and appetite loss. However, this decreases the efficacy of lenvatinib and possibly causes tumor regrowth. The weekend-off protocol, which was defined as a scheduled protocol with administration for a period of five days on and two days off, was the one of the methods for improving tolerability and sustaining efficacy [41]. Although no medical agents relieving these symptoms have been established, one study showed that lenvatinib-related fatigue was associated with carnitine insufficiency [42]. Thus, carnitine supplementation may be effective in improving fatigue in patients receiving lenvatinib. The efficacy of other medical agents, including dexamethasone and Chinese herbal medicines, remains unknown.

While older patients seem to experience more AEs, a retrospective study showed that there were no significant differences in the incidence or severity of AEs between elderly (≥75 years old) and non-elderly patients [43]. Lenvatinib may be safe for elderly patients; however, the study population was relatively small. Incidentally, few reports have investigated whether sex differences or the presence of comorbidity have an impact on AEs.

### 3.2. The Hemorrhage Events

Vascular endothelial growth factor (VEGF) inhibitor treatment was found to be associated with an increased incidence of hemorrhage events, according to a meta-analysis [44,45]; the major frequent hemorrhage events were reported to be upper gastrointestinal hemorrhage and intracranial hemorrhage [46]. Indeed, intracranial hemorrhage was found in three patients that were treated with lenvatinib in the REFLECT trial [12]. Moreover, intraperitoneal or intratumoral hemorrhage is a rare but life-threatening complication. As the original nature, HCC tends to cause intraperitoneal or intratumoral hemorrhage spontaneously. Although there is geographic variation in the incidence of spontaneous HCC rupture, it is reported to be 2.6% in Western countries and 10–26% in Asian countries [47]. Spontaneous tumor rupture was found in approximately 2.6% patients according to the latest nationwide survey of Japan [48]. Accordingly, lenvatinib treatment was deemed to increase the risk of tumor hemorrhage. While the incidence of tumor hemorrhage during lenvatinib treatment remains unknown, Uchida-Kobayashi et al. [49] reported that the lenvatinib-induced tumor hemorrhage was found in five patients with a median onset of three days after the initiation of lenvatinib. The median maximum tumor diameter was 10 cm in these five patients, which is in agreement with the risk factors for spontaneous HCC bleeding [50]. Hence, careful monitoring is required when administering lenvatinib to patients with huge HCC and further research is warranted in order to clarify the clinical features of tumor hemorrhage during lenvatinib treatment.

### 3.3. Hepatic Encephalopathy and Ascites

Hepatic encephalopathy rarely occurred during lenvatinib treatment. The risk factors for hepatic encephalopathy were shown to be an elevated blood concentration of ammonia at baseline and in the presence of portosystemic shunt [22]. The mechanism has remained unclear. One hypothesis is that VEGF inhibitors, including lenvatinib, increase intrahepatic vascular resistance and reduce intrahepatic blood flow via a reduction in nitric oxide production [51]. In fact, the portal venous flow velocity that was assessed by Doppler ultrasonography during lenvatinib treatment was significantly reduced in comparison to baseline [52,53]. Accordingly, the administration of lenvatinib aggravated the portal hypertension and increased the blood flow in portosystemic shunt, which resulted in the occurrence of hepatic encephalopathy. In this connection, a retrospective study revealed that hepatic ascites was found in approximately 29% patients during lenvatinib treatment and that Child-Pugh score 6 and a low platelet count (<12 × 10^4^/μL) were significant risk factors [35]. Because the patients with a low platelet count tend to have portal hypertension, lenvatinib was presumed to have an impact on the further increase in portal hypertension, which resulted in the appearance of hepatic ascites.

### 3.4. Thyroid Dysfunction

Thyroid dysfunction is a frequent complication in patients treated with tyrosine kinase inhibitors (TKIs). Sunitinib and pazopanib, which are oral multi-TKIs, caused treatment-related hypothyroidism in approximately 24% and 12% patients, respectively, according to the phase 3 trial in renal cell carcinoma [54]. In the REFLECT trial, hypothyroidism was found in 78 (16%) and eight (2%) patients who received lenvatinib and sorafenib, respectively [12]. While the clear mechanism underlying the development of thyroid dysfunction due to TKIs remains uncertain, several possible mechanisms have been proposed. One plausible mechanism is that the TKIs cause the destructive thyroiditis [55]. Another mechanism is that TKIs inhibited the binding of VEGF to normal thyroid cells and/or reduced the thyroid flow, which resulted in thyroiditis [55].

According to a retrospective study that included 50 lenvatinib-treated patients, subclinical hypothyroidism, overt hypothyroidism, and thyrotoxicosis occurred in seven (14.0%), 26 (52.0%), and five (10.0%) patients, respectively [56]. PFS in patients with hypothyroidism was longer than that in those without hypothyroidism. However, one retrospective study showed that grade 1 and 2 thyroid dysfunction occurred in 6 (7.8%) and 14 (18.2%) patients, respectively, and grade 2 thyroid dysfunction was found to be a significantly unfavorable factor for PFS in a multivariate analysis [23]. On the other hand, another study showed that lenvatinib-induced hypothyroidism of grade 1, 2, and 3 was found in six (13%), 12 (26%), and one (2%) patients, respectively, and grade 2/3 hypothyroidism was found to be a favorable factor that affected OS in a multivariate analysis [57]. Thus, based on the obtained results of these studies, the relevance of thyroid dysfunction to the clinical outcomes remains controversial.

## 4. Lenvatinib for Patients with Barcelona Clinic Liver Cancer Intermediate Stage

### 4.1. Transcatheter Arterial Chemoembolization and Heterogeneity

Transcatheter arterial chemoembolization significantly improved OS in patients with unresectable HCC in comparison to a placebo group [58,59]. It has become the standard treatment for patients with BCLC intermediate stage and it is widely used in the clinical setting [2]. However, various molecular targeted agents (MTAs) have been available for advanced HCC and the treatment strategy for BCLC intermediate stage will be reconsidered.

According to a systemic review of 10,108 patients that were treated with lipiodol-based TACE, the survival rate at one-, two-, three-, and five-year was reported to be 70.3%, 51.8%, 40.4%, and 32.4%, respectively, with a median OS of 19.4 months [60]. However, not all patients can obtain benefits from TACE [61]. The heterogeneity in BCLC intermediate stage has led to the development of several prognostic scores that are based on liver function and tumor burden that attempt to determine who might benefit most [62]. To date, Bolondi’s subclassification [63], hepatoma arterial embolization prognostic (HAP) score [64], and the Kinki criteria [65] were proposed. Based on these subclassification, TACE was recommended in either patients with low tumor burden or those with good liver function, or both.

A retrospective study analyzed the clinical outcomes in treatment-naïve patients who initially received TACE treatment, and indicated that patients who achieved an objective response with initial TACE showed the longest survival, followed by patients who subsequently obtained an objective response after at least two sessions and those who did not achieve an objective response during the course of treatment [66]. This study also showed that large (>5 cm) and multiple (>4) tumors were independently associated with a failure to achieve a complete response [66]. The response rate of additional TACE reduced in comparison to that of initial TACE [67]. Moreover, additional TACE increased the risk of deterioration of the liver function. Therefore, repeated TACE was not recommended based on its association with the deterioration of the liver function and a poor therapeutic response.

### 4.2. Transcatheter Arterial Chemoembolization Refractoriness

During the sorafenib era, TACE failure/refractoriness was proposed as a criterion for switching from TACE to systemic therapy [3]. The TACE failure/refractoriness was defined as the following characteristics: (1) two or more consecutive insufficient responses of the treated tumor (viable lesion > 50%); and, (2) two or more consecutive progressions in the liver (tumor number increases when compared to tumor number before the previous TACE procedure) [3]. Two retrospective studies [68,69] demonstrated that switching to sorafenib treatment was associated with better survival in comparison to repeated TACE. A prospective international observational trial (OPTIMIS trial) showed that, in TACE-refractory patients, the survival time was better in those who received sorafenib in comparison to those who continued TACE [70]. In addition, a recent study showed that the median PFS in TACE-refractory patients that were treated with lenvatinib, sorafenib, and TACE was 5.8, 3.2, and 2.4 months, respectively [71]. Our retrospective studies reported that lenvatinib treatment was associated with a high ORR and good PFS in patients with intermediate stage HCC [18]. Therefore, lenvatinib treatment could result in a good outcome in TACE-refractory intermediate stage HCC patients.

### 4.3. Exceeding the up-to Seven Criteria, Transcatheter Arterial Chemoembolization Unsuitability and Upfront Systemic Therapies

The up-to seven criteria were defined as the sum of the maximum tumor diameter in the liver (cm) and the number of tumors and they were originally developed for liver transplantation [72]. In patients exceeding the up-to seven criteria, TACE treatment was reported to be likely to worsen the liver function, which may result in missing the chance to receive MTA treatment [73,74]. The proof of concept study demonstrated that, among intermediate stage HCC patients who exceeded the up-to seven criteria, the lenvatinib group had a higher ORR and longer PFS and OS than the TACE group [75]. When the ALBI scores at baseline and at the end of treatment were compared, the ALBI score was sustained in the lenvatinib group, while it worsened in the TACE group [75]. The results of this study demonstrated that lenvatinib is superior to TACE in intermediate stage HCC patients exceeding the up-to seven criteria. The Asia-Pacific Primary Liver Cancer Expert (APPLE) Consensus Statement proposed that the TACE unsuitability should be defined by the following characteristics: (1) unlikely to respond to TACE (confluent multinodular type, massive or infiltrative type, simple nodular type with extranodular growth, poorly differentiated type, intrahepatic multiple disseminated nodules, or sarcomatous changes after TACE); (2) likely to develop TACE failure/refractoriness (exceeding the up-to seven criteria); and, (3) likely to become Child-Pugh B or C after TACE (exceeding the up-to seven criteria and mALBI grade 2b) [76]. The AASLD Consensus Conference showed that locoregional TACE may still be best approach when patients have a low tumor burden and nodules accessible super-selectively [77]. In contrast, upfront systemic therapy may be better for the patients exceeding the up-to seven criteria [77].

Considering these previous reports, when patients have a low tumor burden, selective TACE will be recommended for obtaining a curative response. Non-selective TACE was not recommended, due to the low rate of curative response and risk of deterioration of the liver function. When patients meet the definitions of TACE refractoriness or TACE unsuitable, switching to systemic therapy, including lenvatinib, may be better for avoiding impairment of the liver function. Upfront lenvatinib may be better in patients with a high tumor burden or who are deemed to be refractory to TACE. When a tumor responds well, additional TACE (or surgical resection or ablation) will be considered. Because lenvatinib rapidly reduced the tumor enhancement on radiological imaging and achieved the high ORR in BCLC intermediate stage, the upfront lenvatinib strategy is a promising treatment. However, the efficacy and safety of this strategy has not been validated. In addition, the optimal timing of additional TACE treatment has not been established. Further research to investigate the efficacy and safety of this strategy will be necessary in the future (Figure 1).

Recently, an RCT comparing the efficacy and safety of TACE plus sorafenib to TACE alone (TACTICS trial) indicated that PFS in a TACE plus sorafenib group was significantly longer than that in a TACE alone group; however, a benefit in OS was not proven [78]. Systemic therapy improves the clinical outcomes of TACE, presumably by promoting vascular normalization and contributing to the denser deposition of lipiodol [76]. Hence, lenvatinib-TACE sequential therapy might be a promising treatment for patients with intermediate HCC.

## 5. Post-Progression Treatment after Lenvatinib

### 5.1. Changes in the Liver Function during Lenvatinib Treatment

Hiraoka et al. compared the ALBI score at baseline, and at two weeks and four weeks of treatment in 123 unresectable HCC patients receiving lenvatinib (44 patients (35.8%) with ALBI grade 1 and 50 patients (40.6%) with BCLC early or intermediate stage), and demonstrated a significant decline in the liver function between baseline and two weeks and between baseline and four weeks [79]. We analyzed the change in the ALBI score and parameters that are associated with the liver function, indicating that the ALBI score and albumin worsened during lenvatinib treatment while the total bilirubin level and prothrombin time were sustained [35]. Male sex, ALBI grade 1, CP-5A, and BCLC early or intermediate stage were the significant favorable factors keeping the liver function of Child-Pugh class A in the multivariate analysis [35]. However, a proof of concept study showed that the ALBI score was maintained at the start of treatment in comparison to at end of treatment in patients with BCLC intermediate stage [75]. This result seems to be quite different from the results that were reported by Hiraoka et al. and our previous study. The causes of this differences were presumed to be due to the differences of patients’ background. This proof of concept study included 63% patients with ALBI grade 1 and all patients with BCLC intermediate stage. In these patients, the liver function was likely to be maintained, based on the results of our previous report [35]. Moreover, we also reported that the liver function was better preserved in patients with CP-5A and ALBI grade 1 than in those with CP-5A and ALBI grade 2 or those with Child-Pugh score 6 [35]. Accordingly, the administration of lenvatinib was favorable for patients with ALBI grade 1 and BCLC intermediate stage in terms of the preservation of the liver function.

### 5.2. The Frequency and Predictive Factors of Transition to Post-Progression Treatment

A question remains as to whether lenvatinib or sorafenib is better for first-line treatment. While lenvatinib showed the better ORR and longer PFS in comparison to sorafenib according to the REFLECT study [12], effective MTAs after lenvatinib have not yet been established [39,80]. On the other hand, regorafenib [14], cabozantinib [15], and ramucirumab (limited to AFP ≥ 400 ng/mL) [16] showed clinical benefit after sorafenib, according to randomized control trials (Figure 2). Post-progression survival was strongly correlated with OS in patients that were treated with sorafenib [81]. In addition, a post-hoc analysis of the REFLECT trial revealed that 156 patients (32.6%) who received any anticancer medication (including sorafenib, fluorouracil, and cisplatin) after lenvatinib treatment showed better OS in comparison to the overall population of patients who received lenvatinib (20.8 months vs. 13.6 months) [82]. While there are no definitive answers to this question at this time, post-progression survival plays an important role in increasing the survival curve and establishment of effective treatment after lenvatinib is urgently need.

Nowadays, Child-Pugh class A and a PS of ≤1 are the key eligibility criteria for second-line MTA treatment. The frequency of transition to post-progression treatment was reported to be 43.8–50.9% [35,83,84] and the predictive factors were ALBI grade 1 [35], mALBI grade 1 or 2a [83,84], and Child-Pugh class A [35] (Table 3). Maintaining the liver function could contribute to increasing the possibility of post-progression treatment, resulting in the achievement of good OS. Indeed, a real-world study showed that sequential therapy with TKIs, including lenvatinib, was associated with a favorable outcome [85]. A retrospective study reported a disease control rate (DCR) of 60% in 10 patients who received ramucirumab treatment after lenvatinib [86]. Another retrospective study showed a DCR of 33% in six patients that were treated with ramucirumab after lenvatinib [87]. The populations of these two studies were too small to draw any definitive conclusions regarding the efficacy of ramucirumab treatment after lenvatinib failure. In addition, the frequency of ramucirumab treatment candidates (defined as Child-Pugh class A, PS ≤ 1, and AFP ≥ 400 ng/mL) was <20% [35,83]. A further study should be undertaken in order to establish effective anti-cancer medications after lenvatinib.

### 5.3. The Comparison of Sorafenib and Lenvatinib Treatment

A question remains whether patients who receive lenvatinib or those who receive sorafenib as first-line treatment have a better likelihood for receiving post-progression treatment. A comparative study using propensity score matching reported that lenvatinib-treated patients showed a maintained or improved Child-Pugh score at four weeks or 12 weeks in comparison to sorafenib-treated patients, and worse Child-Pugh score at four weeks contributed to unfavorable OS [88]. Another study showed that the percentage of patients who received secondary treatment in the lenvatinib group was lower than that in the sorafenib group [89]. The post-hoc analysis of a phase 3 study showed that the frequency of receiving anticancer medication did not differ to a statistically significant extent between lenvatinib- and sorafenib-treated patients [82]. Accordingly, no conclusive findings can be drawn from these previous studies.

## 6. Nutrition Assessment and Sarcopenia

### 6.1. Nutrition Assessment as a Prognostic Indicator

Most of the patients with HCC had underlying chronic liver disease and many developed cirrhosis. A high prevalence of malnutrition and protein depletion were observed in cirrhosis patients [90]. Accordingly, it is important to clarify the relationship between nutrition assessment and the prognosis in HCC patients. Some studies have reported the usefulness of nutrition assessment in patients treated with lenvatinib. The neutrophil-to-lymphocyte ratio, which was calculated as the absolute neutrophil count divided by the absolute lymphocyte count and was reported to be a prognostic biomarker in HCC patients treated with sorafenib [91], was also shown to be associated with the clinical outcome in lenvatinib-treated HCC patients [92]. They also reported that the platelet-to-lymphocyte ratio at baseline predicted OS in patients with unresectable HCC who received lenvatinib [93]. The controlling nutritional status (CONUT) score, which is a useful tool for assessing immunonutrition, was calculated based on the serum level albumin, the total peripheral blood lymphocyte count, and the total cholesterol level [94]. The duration of lenvatinib treatment and overall survival were significantly longer in patients with low CONUT score (<5) than those in patients with a high CONUT score (≥5) [95]. The prognostic nutritional index (PNI), which was proposed by Onodera, is calculated based on the serum albumin level and the total peripheral lymphocyte count. It is reported to be a useful tool for assessing the preoperative condition and outcome of patients with gastrointestinal tumors [96]. Some studies have reported that the PNI is a prognostic factor in HCC patients receiving surgical treatment [97,98] or sorafenib treatment [99,100]. The PNI was strongly correlated with the ALBI score and the predictive cut-off values of the PNI for ALBI grade 1 and mALBI grade 2a were 39.0 and 36.0, respectively [101]. Based on these studies, the nutrition status is considered to impact the prognosis in advanced HCC patients who are treated with lenvatinib. However, which of these nutrition assessments is the better prognostic predictor remains unclear, due to a lack of comparative studies (Table 4).

### 6.2. The Relationship between Sarcopenia and Overall Survival

Sarcopenia is defined as the reduced the muscle mass and strength [102]. Skeletal muscle mass is not only a good indicator of nutrition in patients with cirrhosis, but it has also been shown to be closely associated with the prognosis of HCC patients [102]. A retrospective study investigated the impact of sarcopenia on the clinical outcome of HCC patients who received lenvatinib treatment, and it indicated that the patients with high skeletal muscle mass (SMI) showed better OS in comparison to those with low SMI [103]. Skeletal muscle mass was measured based on computed tomography images and it was normalized for height in m^2^ as the SMI. The time to treatment failure (defined as the period between the initiation of lenvatinib and either the end of lenvatinib treatment or the end of the analysis) was significantly longer in patients with high SMI values than in those with low SMI values. AEs that resulted in discontinuation were less frequently found in patients with high SMI values than in those with low SMI values. Another retrospective study showed that the OS of patients with a normal grip strength (GS) was better than that in those with decreased GS, and a normal GS was found to be significantly associated with OS in a multivariate analysis [104]. While the ORR and PFS did not differ to a statistically significant extent between the two groups, the post-progression survival in patients with a normal GS was longer in comparison to that in patients with decreased GS, which indicated that patients with a normal GS may be more likely to have a maintained liver function and PS, raising the possibility of anticancer therapy after progression [104]. Although studies concerning sarcopenia have been limited, Sarcopenia might impact the therapeutic efficacy of lenvatinib treatment. Further studies were warranted for investigating the role of sarcopenia in HCC patients who are treated with lenvatinib (Table 4).

## 7. Lenvatinib plus Pembrolizumab Therapy

Immune checkpoint inhibitors monotherapy has been reported to be associated with promising clinical outcomes in HCC patients. A phase 1/2 study of nivolumab in advanced HCC patients in the dose expansion phase showed an ORR of 20% with a median duration of response of 9.9 months [105]. A phase 2 trial of pembrolizumab enrolled 104 patients with Child-Pugh class A and a history of sorafenib treatment, and reported an ORR of 17% [106]. The median duration of response was not reached at the time of the analysis (range, 3.1–14.6 months) [106]. A phase 1/2 study of nivolumab plus ipilimumab reported an ORR of 32% in patients who were previously treated with sorafenib in arm A (nivolumab (1 mg/kg) plus ipilimumab (3 mg/kg every three weeks; four doses) followed by nivolumab (240 mg, intravenously, every two weeks) [107]. Based on these studies, the US Food and Drug Administration (FDA) approved nivolumab, pembrolizumab, and combination therapy with nivolumab plus ipilimumab for use as second-line treatment after sorafenib treatment.

The hypoxic tumor microenvironment makes cancer cells escape the immune surveillance and reduces the function of resident and transiting immune effector cells [80]. Excessive VEGF production (that is principally driven by hypoxia) can express immunosuppressive effects in tumors by inhibiting maturation of dendritic cells and priming subsets of immunosuppressive inflammatory cells [80]. Accordingly, investigators may hypothesize that concomitant VEGF inhibitor would synergistically improve the clinical outcome of immunotherapies [80]. Recently, combination therapy with atezolizumab plus bevacizumab, which inhibits the programmed death-ligand 1 and VEGF pathway, was reported to be superior to sorafenib [108] and it will be used as a first-line treatment. Accordingly, immune checkpoint inhibitors will gain an important position in the treatment of HCC. A phase 1b study of lenvatinib plus pembrolizumab showed an ORR of 46%, as assessed by mRECIST and 36%, as assessed by RECIST ver.1.1 [109]. The median PFS, as assessed by mRECIST and RECIST ver.1.1, was 9.3 months (95% CI 5.6–9.7) and 8.6 months (95% CI 7.1–9.7), respectively, while the median OS was 22.0 (20.4—not estimable) months [109]. These results of the phase 1b study of lenvatinib plus pembrolizumab demonstrated promising antitumor effects in advanced HCC. A phase 3 study (LEAP-002) is currently ongoing (ClinicalTrials.gov Identifier: NCT03713593).

## 8. Efficacy and Safety in Patients Excluded from the REFLECT Study

Patients with the experience of systemic therapy, portal vein invasion at the main portal trunk, bile duct invasion, liver occupation of tumor ≥ 50%, and Child-Pugh class B were excluded in the REFLECT study [12]. Therefore, the therapeutic efficacy of lenvatinib for these patients remains uncertain. Maruta et al. [22] reported that the ORR in patients with and without the experience of systemic therapy was 35% and 44%, respectively. They also reported that the median PFS in patients with Child-Pugh class A who received lenvatinib as first-line treatment was 5.2 months, while that in patients who received lenvatinib as second-line treatment or beyond was 4.8 months [22]. The patients with a high burden of intrahepatic lesions (defined as main portal vein and/or bile duct invasion or occupation of ≥50% of the liver; *n* = 27) were reported to show an ORR of 41%, which was equivalent to the ORR of 41% in patients without a high burden of intrahepatic lesion (*n* = 125) [22]. The PFS in patients with and without high burden of intrahepatic lesion was 3.9 months and 5.3 months, respectively; the difference was not statistically significant [22]. According to a previous study that was reported by Sho et al. [20], the ORR and PFS were not signficantly differed between the patients who met the REFLECT criteria and those who did not (61.5%, 10.3 months vs. 48.3%, 9.8 months). They also reported that the ORR and PFS did not differ to a statistically significant extent between patients with an experience of TKI therapy and those without (44.4% 11.1 months vs. 56.9%, 9.1 months), the patients with portal vein invasion at the main portal trunk and those without (40.0%, 5.6 months vs. 54.3%, 9.9 months), or the patients with liver occupation of tumor ≥50% and those without (33.3%, 8.6 months vs. 56.3%, 9.9 months) [20]. While these studies reported that the efficacy in patients with Child-Pugh class B was not significantly lower in comparison to that in patients with Child-Pugh class A [20,22], one study reported that the ORR in patients with Child-Pugh class B was lower than that in patients with Child-Pugh class A (16.3% vs. 36.5%, *p* = 0.008) [21].

Accordingly, lenvatinib was thought to be effective for patients who were excluded according to tumor conditions; however, the number of subjects in these studies was small. Regarding the treatment settings of lenvatinib, the therapeutic efficacy in patients who received lenvatinib as first-line treatment did not differ to a statistically significant extent from that in patients who received lenvatinib as second-line treatment or beyond [20,22,110]. Based on these findings, the administration of lenvatinib as a second-line treatment or beyond may not be inferior to the administration of lenvatinib as first-line treatment. The efficacy of lenvatinib in patients with Child-Pugh class B has not been determined.

## 9. Conclusions

Lenvatinib was associated with a good therapeutic response in patients with advanced HCC. A well-preserved liver function and BCLC intermediate stage were key factors in achieving therapeutic efficacy. The management of AEs plays an important role in the continuation of lenvatinib treatment. While many MTAs have become available, effective treatments that can be administered after lenvatinib remain to be established. Accordingly, the establishment of effective treatment after lenvatinib is urgently needed. Regarding the treatment strategy for BCLC intermediate stage, upfront lenvatinib, followed by TACE, was a rational treatment approach, as it achieved a good therapeutic response and preserved the liver function, especially in patients with a high tumor burden. In addition, sequential therapy with lenvatinib–TACE might be a promising treatment for patients with intermediate HCC. The nutrition status and sarcopenia were found to be prognostic factors in lenvatinib-treated patients. Lenvatinib was also possibly effective and safe for patients who were excluded from a phase 3 study. Recently, atezolizumab plus bevacizumab has become available for first-line treatment, and it remains unclear whether atezolizumab and bevacizumab were superior to lenvatinib, because of a lack of RCT comparing lenvatinib and atezolizumab combined with bevacizumab. Nonetheless, lenvatinib was adapted as a powerful TKI for advanced HCC. It is hoped that this review will contribute to a better understanding of lenvatinib treatment.

## Figures and Tables

**Figure 1 pharmaceuticals-14-00036-f001:**
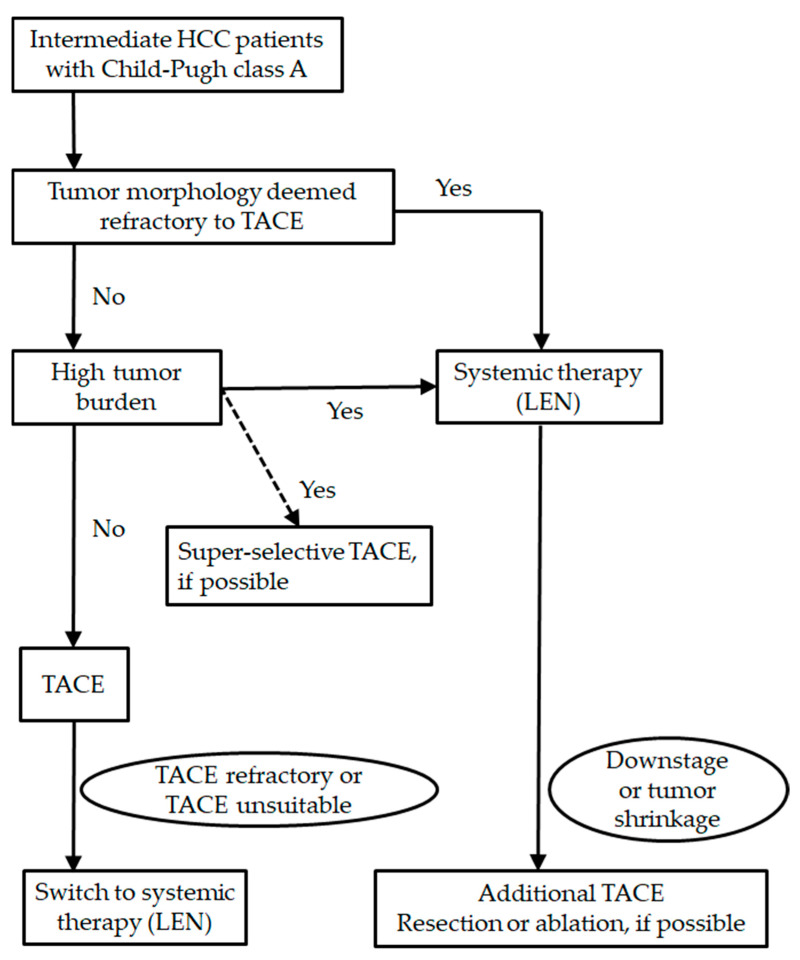
Treatment strategy for Barcelona Clinic Liver Cancer (BCLC) intermediate stage.

**Figure 2 pharmaceuticals-14-00036-f002:**
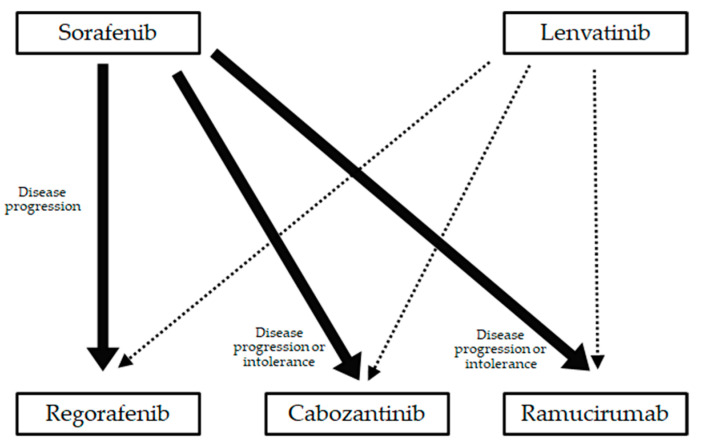
Post-progression treatment. The effective molecular targeted agents after the lenvatinib has not been established yet.

**Table 1 pharmaceuticals-14-00036-t001:** The therapeutic response to lenvatinib treatment.

Author/Ref. No.	Years	Country	No. of pts	ORR (%)		Median PFS (Months)	Median OS (Months)	Predictive Factors of ORR	Commentary
				mRECIST	RECIST ver.1.1				
Kudo M./[12]	2018	Global	478	40.6	18.8	7.3	13.6	NA	REFLECT trial
Ueshima K./[17]	2019	Japan	82	39.0	NA	7.6	could not be reached	ALBI grade 1	ALBI grade 1; ORR of 57.1%, median PFS of 18.9 months
Hatanaka T./[18]	2020	Japan	94	30.4	NA	5.4	NA	BCLC intermediate stage	BCLC intermediate stage; ORR of 47.6%, median PFS 8.0 months
Sasaki R./[19]	2019	Japan	81	34.6	17.3	NA	11.6	NA	The patients with the good ORR and those with high RDI had significantly good OS.
Sho T./[20]	2020	Japan	105	53.5	NA	9.8	NA	NA	
Ogushi K./[21]	2020	Japan	181	30.4	NA	NA	369 (days)	CP-A, PS 0, RDI > 0.70	CP-A; ORR of 36.5%. RDI > 0.70: ORR of 47.7%. CP-A and BCLC intermediate stage were the predictive factors affecting the OS.
Maruta S./[22]	2020	Japan	152	40.8	15.8	5.1	13.3	NA	
Ohki T./[23]	2020	Japan	77	29.9	NA	5.6	NA	CP-A, RDI > 70% at 30 days, AFP reduction	RDI > 70%; ORR of 45.2%, RDI ≤ 70%; ORR of 11.4%. The significant factors associated with the PFS were CP-5A, tumor size ≥ 40 mm among pretreatment factors.
Hiraoka A./[24]	2019	Japan	152	38.7	NA	NA	could not be reached	NA	mALBI 2b or 3 was unfavorable predictive factor relevant to the OS.
Wang D.X./[25]	2020	China	54	NA	22.2	5.6	could not be reached	could not be found in a multivariate analysis.	CP-A, BCLC intermediate stage and PVTT significantly affected the PFS.
Cheon J./[26]	2020	Korea	92	NA	14.1	4.3	7.1	NA	

AFP; α-fetoprotein, ALBI; albumin-bilirubin grade, BCLC; Barcelona Clinic Liver Cancer, CP-A; Child-Pugh class A, mRECIST; modified Response Evaluation Criteria in Solid Tumors, NA; not available, No.; number, ORR; objective response rate, OS; overall survival, PFS; progression-free survival, pts; patients, PVTT; portal vein tumor thrombosis, RDI; relative dose intensity.

**Table 2 pharmaceuticals-14-00036-t002:** Reports associated with dose intensity.

Author/Ref. No.	Years	No. of pts	Assessment	Time	Cut-off Value	Associated Factors	Therapeutic Response	Commentary
Sasaki R./[19]	2019	81	RDI	8 weeks	67%	BMI, PS, BCLC stage, platelet count, PT, albumin, initial dose	OS	Median OS could not be reached during the observation.
Ogushi K./[21]	2020	181	RDI	8 weeks	70%	NA	ORR	Mean RDI gradually decreased from the start of lenvatinib to 2 months.
Ohki T./[23]	2020	77	RDI	30 days	70%	NA	ORR, PFS	RDI > 70%; median PFS of 9.3 months
Kirino S./[28]	2020	60	RDI	4 weeks	70%	Body weight, ALBI score, albumin, grade 3 or 4 adverse events	OS, time to discontinuation of treatment	Median OS could not be reached and median duration of lenvatinib was 9.5 months in patients with RDI > 70%.
Ono A./[29]	2020	41	RDI	4 weeks	70%	Platelet count, albumin, AST, AFP-L3, DCP, ALBI score, TKI experience	PFS	RDI > 70%; median PFS of 6.7 months. RDI in 3–4 weeks significantly reduced compared to 1–2 weeks.
Takahashi A./[30]	2019	50	RDI	8 weeks	75%	Child-Pugh class A, ALBI, EHS, initial dose	ORR, PFS	RDI > 75%; ORR of 68%, median PFS of 7.4 months
Eso Y./[31]	2019	49	DBR, RDI	60 days	DBR of 238.9, RDI of 66%	BSA, mALBI grade 1 + 2a, BTR	ORR, PFS	DBR was calculated as the delivered DI divided by BSA.

AFP-L3; Lens culinaris agglutinin-reactive fraction of alpha-fetoprotein, ALBI score; albumin-bilirubin score, AST; aspartate transaminase, BCLC stage; Barcelona Clinic Liver Cancer stage, BMI; body mass index, BSA; body surface area, BTR; branched-chain amino acids tyrosine ratio, CONUT; Controlling Nutrition Status, DBR; the delivered dose intensity/body surface area ratio, DCP; des-carboxy prothrombin, DI; dose intensity, EHS; extrahepatic spread, NA; not available, No.; number, OS; overall survival, PFS; progression-free survival, PS; performance status, PT; prothrombin time, pts; patients, RDI; relative dose intensity, TKI; tyrosine kinase inhibitor.

**Table 3 pharmaceuticals-14-00036-t003:** Previous studies concerning the candidate for post-progression treatment and ramucirumab treatment.

Author/Ref. No.	Years	No. of pts	Frequency of Post-Progression Treatment	Frequency of Ramucirumab Treatment	Favorable Predictive Factors of Post-Progression Treatment
Hatanaka T./[35]	2020	79	45.6%	13.9%	ALBI grade 1, Child-Pugh score 5
Hiraoka A./[83]	2020	73	43.8%	20.0%	mALBI grade 1 + 2a
Ando Y./[84]	2020	53	50.9%	NA	mALBI grade 1 + 2a

Post-progression treatment was defined as Child-Pugh class A and PS of ≤1, and ramucirumab treatment as Child-Pugh class A, PS of ≤1 and AFP ≥ 400 ng/mL. AFP; α-fetoprotein, ALBI grade; albumin-bilirubin grade, mALBI grade; modified albumin-bilirubin grade, NA; not available, No.; number, pts; patients. PS; performance status.

**Table 4 pharmaceuticals-14-00036-t004:** Previous reports associated with nutrition status and sarcopenia.

Author/Ref. No.	Years	No. of pts	Assessment	Cut-off Value
Tada T./[92]	2020	237	NLR	4
Tada T./[93]	2020	283	PLR	150
Shimose S./[95]	2020	164	CONUT score	5
Hiraoka A./[101]	2020	375	PNI	40
Uojima H./[103]	2020	100	SMI	42 and 38 cm^2^/m^2^ in men and women, respectively
Endo K./[104]	2020	69	grip strength	26 kg and 18 kg in men and women, respectively

CONUT; controlling nutritional status, NLR; neutrophil-to-lymphocyte ratio, No.; number, PLR; platelet-to-lymphocyte ratio, PNI; prognostic nutritional index, pts; patients, SMI; skeletal muscle index.

## Data Availability

No new data were created or analyzed in this study. Data sharing is not applicable to this article.

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
