# Peer review of "Lenvatinib for Hepatocellular Carcinoma: A Literature Review"

_pharmaceuticals, 2021, doi:10.3390/ph14010036_

Round 1

Reviewer 1 Report

Thank you for submitting in Pharmaceuticals.

Lenvatinib, which is an oral multi-kinase inhibitor, has been widely used for advanced hepatocellular carcinoma (HCC). This article reviewed the therapeutic response, management of adverse events, post-progression treatment after lenvatinib and treatment strategy for intermediate stage HCC.

Major problem.

Many reviews have already been reported that discuss the efficacy and safety of Lenvatinib for HCC. However, the authors added value to this treatise by considering the timing of switching from TACE in BCLC intermediate stage HCC and discussing treatment strategy after lenvatinib treatment. Unfortunately, these valuable items are treated like any other mundane item in this manuscript. Therefore, the authors should further discuss and emphasize these valuable items.

Miner problems.

  1. p2, l61. According to--- The argument is suddenly changing. The problems with TACE should be discussed before the usefulness of lenvatinib.
  2. p3, l110. The PFS and OS. The author is too particular about his own treatise. Please discuss it more objectively.
  3. p6, l156. The patients who were excluded from the REFLECT study. This section is confusing. If necessary, why not add it at the end?
  4. p9, l315. The comparison of sorafenib and lenvatinib treatment. This paragraph is confusing. First, the safety and efficacy of Lenvatinib against TACE refractory should be discussed, and then the treatment strategy after Lenvatinib should be discussed.
  5. p10, l326. Nutrition assessment and sarcopenia. Similarly, the priorities are different.

Author Response

Reviewer 1

Major problem.

Many reviews have already been reported that discuss the efficacy and safety of Lenvatinib for HCC. However, the authors added value to this treatise by considering the timing of switching from TACE in BCLC intermediate stage HCC and discussing treatment strategy after lenvatinib treatment. Unfortunately, these valuable items are treated like any other mundane item in this manuscript. Therefore, the authors should further discuss and emphasize these valuable items.

Thank you for your comments.

We moved this paragraph forward and discussed about and emphasize the intermediate stage HCC. We further discuss about the TACE refractoriness (page 8 line 261-270) and added lenvatinib-TACE sequential therapy in page 9 line 312-317

Miner problems.

p2, l61. According to--- The argument is suddenly changing. The problems with TACE should be discussed before the usefulness of lenvatinib.

Thank you for your comments.

We wrote about the TACE treatment and the problem with TACE before the lenvatinib in Page 1 line 40-41

p3, l110. The PFS and OS. The author is too particular about his own treatise. Please discuss it more objectively.

Thank you for your comments.

As reviewer 3 suggested, we described about treatment naïve and extrahepatic spread. We added the sentences about the other studies in line 124-129, making this paragraph more objectively.

p6, l156. The patients who were excluded from the REFLECT study. This section is confusing. If necessary, why not add it at the end?

Thank you for your comments.

We moved it at the end in paragraph of ‘8. Efficacy and safety in patients excluded from the REFLECT study’.

p9, l315. The comparison of sorafenib and lenvatinib treatment. This paragraph is confusing. First, the safety and efficacy of Lenvatinib against TACE refractory should be discussed, and then the treatment strategy after Lenvatinib should be discussed.

Thank you for your comments.

We move the paragraph of ‘Lenvatinib for patients with BCLC intermediate stage’ forward, followed the paragraph of ‘Post-progression treatment after lenvatinib’ (including ‘the comparison of sorafenib and lenvatinib treatment’)

p10, l326. Nutrition assessment and sarcopenia. Similarly, the priorities are different.

Thank you for your comments.

We moved it as mentioned above.

Reviewer 2 Report

Overall comments:

  1. The work by Hatanaka et al. titled ‘Lenvatinib for hepatocellular carcinoma: Literature review’ is a very significant work and can be considered for publication only after careful revision.
  2. Extensive English editing is required.
  3. There could have been more figures and tables to better understand in an easy way
  4. Page 8 line 246: thyroid …should be Thyroid. Please double check all the punctuation missing throughout the manuscript.
  5. There are some reviews already published in the same topics this year. Therefore, authors are suggested to rewrite the review in as a comprehensive review and that could be considered for publication.

Specific comments:

Page 3 line 111: 7.3 months…please make it clear.  

Page 3 line 113: Authors reported that…which authors reported?

Page 6 line 157: As authors mentioned above…who?

Author Response

Reviewer 2.

Overall comments:

  1. The work by Hatanaka et al. titled ‘Lenvatinib for hepatocellular carcinoma: Literature review’ is a very significant work and can be considered for publication only after careful revision.
  2. Extensive English editing is required.

Thank you for your comments.

Our manuscript received English editing and we changed the manuscript clearly.

  1. There could have been more figures and tables to better understand in an easy way

Thank you for your comments.

We added the Figure 2 and Table 4.

  1. Page 8 line 246: thyroid …should be Thyroid. Please double check all the punctuation missing throughout the manuscript.

Thank you for your comments.

We changed the ‘Thyroid dysfunction’ and checked all manuscript.

  1. There are some reviews already published in the same topics this year. Therefore, authors are suggested to rewrite the review in as a comprehensive review and that could be considered for publication.

Thank you for your comments.

As Reviewer 1 also suggested, treatment strategy for BCLC intermediate stage is hot topic and an important issue. Therefore, we further discussed about the TACE refractoriness (page 8 line 261-270) and added lenvatinib-TACE sequential therapy in page 9 line 312-317, making our manuscript more comprehensive.

Specific comments:

Page 3 line 111: 7.3 months…please make it clear.

Thank you for your comments.

We changed this sentence as ‘The median PFS in lenvatinib-treated patients was reported to be 7.3 months (95% confidence interval [CI] 5.6-7.5) in a phase 3 study’ in page 3 line 115-116.

Page 3 line 113: Authors reported that…which authors reported?

Thank you for your comments.

We changed ‘Authors’ to ‘our retrospective study’.

Page 6 line 157: As authors mentioned above…who?

Thank you for your comments.

This is confusing. So, we deleted ‘As authors mentioned above’.

Reviewer 3 Report

The present manuscript by Hatanaka et al., Reviewed the therapeutic response, management of adverse events, post-progression after Lenvatinib therapy in hepatocellular carcinoma (HCC)

Thanks to the authors for the effort because HCC is a global burden the most we know, the better in order to develop an effective treatment.

I have some comments about the manuscript:

Lenvatinib versus Sorafenib: What are the differences in the mechanism of action?

The presentation is not very careful.

Tables cannot be read in the present format

An extensive English edition is needed

The conclusion section is too short and un-explanatory

Which are the advantages and inconvenience to use Lenvatinib instead of other drugs?

Please describe figure 1

How about age, sex, or comorbidity of patients ...? Did these parameters influence the result?

Have treated patients with recurrent tumors or metastasis? If yes, is there a difference in the effect?

Is hepatocellular carcinoma the primary or is it due to metastasis? what about the effect of Lenvatinib in these cases?

Author Response

Reviewer 3.

The present manuscript by Hatanaka et al., Reviewed the therapeutic response, management of adverse events, post-progression after Lenvatinib therapy in hepatocellular carcinoma (HCC)

Thanks to the authors for the effort because HCC is a global burden the most we know, the better in order to develop an effective treatment.

I have some comments about the manuscript:

・Lenvatinib versus Sorafenib: What are the differences in the mechanism of action?

Thank you for your comments.

Sorafenib inhibits tyrosine kinase receptor including VEGF receptors (VEGFRs) and PDGF receptor-β and drivers of cell proliferation such as RAF1, BRAF, and KIT. Lenvatinib acts as an inhibitor of VEGFR 1–3, fibroblast growth factor receptors 1–4, PDGF receptor-α, KIT, and RET.

We added the mechanism of sorafenib in line 46-48.

・The presentation is not very careful.

We apologize for being hard to read our manuscript. Our manuscript received English editing and we changed the manuscript clearly.

・Tables cannot be read in the present format

We apologize for being inconvenience.

We made the tables more clearly.

・An extensive English edition is needed

Thank you for your comments.

Our manuscript received English editing and we changed the manuscript clearly.

・The conclusion section is too short and un-explanatory

Thank you for your comments.

We added the information in page 15 line 496-502.

・Which are the advantages and inconvenience to use Lenvatinib instead of other drugs?

Please describe figure 1

Thank you for your comments.

While lenvatinib showed the better ORR and longer PFS than sorafenib according to the REFLECT study, the effective molecular targeted agents (MTA) after the lenvatinib has not been established yet. On the other hand, regorafenib, cabozantinib and ramucirumab (limited to AFP≥400ng/ml) showed the clinical benefit after the sorafenib according to the randomized control trials.

We discussed it in page 9 line 342-353 and made the Figure 2.

・How about age, sex, or comorbidity of patients ...? Did these parameters influence the result?

Thank you for your comments.

A retrospective study showed that there were no significant differences in the incidence and severity of AE between the elderly (≥75 years old) and non-elderly patients. Lenvatinib might be used for safely for the elderly patients although the subjects enrolled in this study was small. Few reports were available for whether sex differences and presence of comorbidity impact the AEs.

We discussed it in page 7 line 195-199.

・Have treated patients with recurrent tumors or metastasis? If yes, is there a difference in the effect?

Thank you for your comments.

There were no differences in PFS between the treatment-naïve patients and patients receiving previous treatment. The PFS in patients with extrahepatic spread were shorter than that in patients without extrahepatic spread.

We discussed it in page 3 line 119-122.

・Is hepatocellular carcinoma the primary or is it due to metastasis? what about the effect of Lenvatinib in these cases?

Thank you for your comments.

The PFS in patients with extrahepatic spread was shorter in comparison to patients without extrahepatic spread

We discussed it in page 3 line 120-122.

Round 2

Reviewer 1 Report

Lenvatinib, which is an oral multi-kinase inhibitor, has been widely used for advanced hepatocellular carcinoma (HCC). This article reviewed the therapeutic response, management of adverse events, post-progression treatment after lenvatinib and treatment strategy for intermediate stage HCC.

Thank you for submitting in Pharmaceuticals.

Many reviews have already been reported that discuss the efficacy and safety of Lenvatinib for HCC. However, the authors added value to this treatise by considering the timing of switching from TACE in BCLC intermediate stage HCC and discussing treatment strategy after lenvatinib treatment. Major problem of this article was these valuable items were treated like any other mundane item in this manuscript. The authors had further discussed and emphasize these valuable items.

Minor problems have also been fixed in the revised version.

Author Response

Reviewer 1

Many reviews have already been reported that discuss the efficacy and safety of Lenvatinib for HCC. However, the authors added value to this treatise by considering the timing of switching from TACE in BCLC intermediate stage HCC and discussing treatment strategy after lenvatinib treatment. Major problem of this article was these valuable items were treated like any other mundane item in this manuscript. The authors had further discussed and emphasize these valuable items.

Minor problems have also been fixed in the revised version.

Thank you for your comment.

We further discuss and emphasize the treatment strategy for intermediate stage. We added in page 8 line 267-275 and page 9 line 288-295

Reviewer 2 Report

The paper could be considered for publication only after the minor revision. 

1.  I think many of the important review and research paper have not been discussed. For example,  DOI: 10.2147/JHC.S168953; DOI: 10.1080/14737140.2018.1524297; DOI: 10.1371/journal.pone.0244370; doi: 10.3747/co.27.7159. 

These above mentioned papers could be discussed. 

2. Ref 62 is an article from Oncotarget. Since 2017 there are a lot of controversy about Oncotarget. Therefore, I suggest to avoid their article to cite. 

Author Response

Reviewer 2

The paper could be considered for publication only after the minor revision.

  1. I think many of the important review and research paper have not been discussed. For example, DOI: 10.2147/JHC.S168953; DOI: 10.1080/14737140.2018.1524297; DOI: 10.1371/journal.pone.0244370; doi: 10.3747/co.27.7159.

These above mentioned papers could be discussed.

Thank you for your comment.

We cited these review and research paper as reference 36, 39, 62 and 80. We discussed in page 3 line 140, page 6 line 172-174, page 8 line 271-273, page 11 line 368, and page 14 line 477-483.

  1. Ref 62 is an article from Oncotarget. Since 2017 there are a lot of controversy about Oncotarget. Therefore, I suggest to avoid their article to cite.

Thank you for your comment.

We deleted reference 62.

Reviewer 3 Report

Thanks to the authors for the revised version of the manuscript Lenvatinib for hepatocellular carcinoma: A literature review.

My major concerns have been mostly solved.
I have only some minor comments:
Dear authors, please take care of the way that you present your work and also be sure that you are following the journal rules
Please, justify the paragraphs
I would prefer no abbreviation in the titles of the subsections
Please define PFS the first time that is mentioned in the text, the same RCT
Please define pt in table 1 and pts in table 2 and revise abreviatuions

Lines 122-124: the text is confusing
Figure 2 is too big

Author Response

Reviewer 3

My major concerns have been mostly solved.

I have only some minor comments:

Dear authors, please take care of the way that you present your work and also be sure that you are following the journal rules

Thank you for your meaningful comments and we changed our manuscript as you pointed out.

Please, justify the paragraphs

Thank you for your comment.

We reduced the space above and below the paragraphs.

I would prefer no abbreviation in the titles of the subsections

Thank you for your comment.

We wrote the full spell in the titles of the subsections and deleted the abbreviation.

Please define PFS the first time that is mentioned in the text, the same RCT

Thank you for your comment.

We added the definition of PFS in page 3 line 117-118 and that of RCT in page 2 line 51-52.

Please define pt in table 1 and pts in table 2 and revise abbreviation

Thank you for your comment.

We defined pt and pts as patients. We added the abbreviation in Table 1-4.

Lines 122-124: the text is confusing

We apologized for inconvenience.

‘that’ is redundant. We delete the ‘that’.

Figure 2 is too big

Thank you for your comment.

We made Figure 2 small.

This manuscript is a resubmission of an earlier submission. The following is a list of the peer review reports and author responses from that submission.